# A Bayesian Approach
# Towards Crowdsourcing the Truths from LLMs

**Peiran Yao[1], Jerin George Mathew[2], Shehraj Singh[1],**
**Donatella Firmani[2], Denilson Barbosa[1]**
[1]University of Alberta [2]Sapienza University of Rome
{peiran,denilson}@ualberta.ca

## Abstract

Concerns persist over the trustworthiness of large language models (LLMs) due to the generation of plausible but incorrect information, known as hallucination. Existing approaches focus on identifying false answers or improving correctness by sampling responses from a single LLM. However, querying multiple LLMs, which exhibit complementary strengths, remains largely unexplored. In this work, we propose a Bayesian crowdsourcing approach towards aggregating multiple answers from multiple LLMs and quantifying their uncertainty. Extending the Dawid-Skene model, we treat LLMs as annotators, using their answer probabilities as noisy observations of truthfulness and modeling semantic relations between answers in the covariance structure, and jointly learn about LLM's reliability and calibration as parameters. Validated across three open-domain question answering dataset, results show that our approach outperforms existing statistical or agentic methods in abstaining from false answers and identifying truthful ones, offering a robust, scalable solution for uncertainty quantification and truth discovery in LLM outputs.

## 1 Introduction

Large language models (LLMs; [29, 7, 40, 15] *iter alia*) are pre-trained on web-scale language data that make them excel at generating human-like responses and storing extensive world knowledge [32]. They have demonstrated remarkable capabilities across a wide range of tasks that require conceptual knowledge, from recalling answers to trivia questions [27] to performing complex multi-hop reasoning [18]. However, their wide applications have raised concerns over the trustworthiness and reliability of their outputs [25, 13], exemplified by the generation of plausible but incorrect information (i.e. "hallucination" [47]) and the lack of self-awareness of limitations in knowledge [17], which both remain largely unaddressed.

To improve the reliability of LLM answers at test time, strategies that work on one of the two complementary fronts have been proposed: improving the correctness of answers [26, 45, 13, 6], or identifying answers that are more likely to be false to abstain from them [9, 20, 24, 8]. The underlying ideas behind strategies from both fronts are similar: they rely on sampling multiple answers from a *single* LLM and aggregating them based on consistency [45, 20, 24], or following an agentic workflow where LLMs verbally evaluate or improve an answer, similarly to human interactions [26, 6, 8].

We study a more **general scenario**, where *multiple* LLMs are each queried *multiple* times to generate a set of candidate answers for *multiple* questions, that to our knowledge has not been extensively and systematically studied. Querying multiple LLMs, rather than a single one, could be helpful because they are known to have different strengths and weaknesses [16, 43], and their outputs can be complementary [42]. The goals are, at the same time, (1) to quantify the uncertainty (**UQ**) of these

Workshop on Bayesian Decision-making and Uncertainty, 38th Conference on Neural Information Processing Systems (NeurIPS 2024).

answers, as a base for abstaining; and (2) to aggregate these answers to infer the truthful answer for each question, a problem known as truth discovery (**TD**) in the data management literature [23, 48].

This task can be seen as a special case of inferring the ground truths from multiple annotators, where the annotators are all LLMs. The base problem, known as crowdsourcing [48], is well-established with Bayesian models that go beyond simple consistency-based aggregation [4, 46], leading to applications such as in NLP tasks [30, 31, 38]. Despite precursory works [5] on crowdsourcing with weak systems as annotators, not much has been done to extend these models to LLM-based annotators. Crowdsourcing models are mainly limited to classification tasks with predefined classes, while the typical use cases of LLMs require free-form text answers. Moreover, classical crowdsourcing models expect a single answer from each annotator, while LLMs can generate multiple answers for a single question with different probabilities, which could additionally inform crowdsourcing models.

To address these challenges, we propose a probabilistic generative model that extends the Dawid-Skene model [4] to the LLM-based setting. For each question, the *truthfulness* of candidate answers from multiple LLMs is modeled as a multivariate latent variable whose correlation structure is tied to the semantic relations between the answers, and whose marginal distribution determined by the *reliability* of the LLMs. The observed data are the probabilities of the candidate answers being generated by the LLMs, which are treated as noisy observations of the truthfulness and are modeled as dependent on the latent truthfulness variable via a *calibration* process. Parameters for reliability and calibration are learned by the EM algorithm, and the truthfulness (inverse of uncertainty) of the candidate answers is inferred by the posterior of the latent variable. We validate our model on three open-domain question answering datasets, where we show that our model can even outperform costly agentic methods in effectively identifying the truthful answers and abstaining from the false ones.

Figure 1: We propose a Bayesian model for crowdsourcing from free-text outputs of multiple LLMs. To better illustrate the relations between elements of the multivariate random variable $y$, this figure does not strictly follow the standard plate notation. $y$ as a single multivariate random variable is represented as a rounded rectangle with its elements represented as circles inside, which should not be confused with repeated independent variables that are conventionally represented as a rectangle.

## 2   Related Work

**Crowdsourcing.**   Combining multiple answers by simple majority voting is commonly used to improve LLMs at test time [45, 22]. Crowdsourcing models such as Dawid-Skene [4] opt for iterative weighted majority voting by assuming different reliability of workers. They are comprehensively reviewed by [48, 31, 30] and implemented by [41]. Typically, these models require a known and fixed set of possible answers, while recent studies [21] operate with free-text by a weighted majority voting in the embedding space. Our approach eliminates the need for embedding and provides more flexibility and interpretability. Weak supervision [36], where the truths are not fully available and truth inference is integrated with model learning, is closely related. We only consider the case when the ground truth is completely unavailable, and do not consider learning. Second-order information such as worker's predictions [19] is useful but beyond the scope of this work.

**Uncertainty quantification.**   Many UQ methods work by quantifying how consistent the answers are, with [20, 24, 34] providing methods for a semantic-aware consistency measure. Picking the most consistent answer is also the underlying strategy to improve the factuality of LLMs [45, 22]. [12] infers about the uncertainty of a statement using a Hidden Markov Model by considering the logical relations with other statements conditionally generated from the initial statement. Our model offers more relaxations and consider arbitrary answers sampled independently from multiple LLMs. A separate step in UQ is to calibrate the uncertainty estimates to match accuracy using labeled

dataset [10], while our model allows the learning of the calibration and reliability parameters without labelling.

**Scaling LLMs at Test Time**   Doing repeated sampling allows for a trade-off between the amount of computing spent in test time and the quality of the final output, a research topic known as *test-time* or *inference-time* scaling [45, 22]. Although self-consistency (simple majority voting) yields good performance [45, 22], it is typically assumed that a verifier trained on the same data is available so that more complex aggregation methods can be used, such as voting weighted by verifier score, or *best-of-n* selected by the verifier [37]. Such a verifier could be available when the ground-truth is known for a subset of the data [3], or when there are deterministic rules to verify the answers such as for math and coding problems [44]. However, it remains an open question to find a general-purpose verifier. Our method works in the most unrestricted setting and assumes that there is no access to any verifier, nor is there information about the relative strengths of LLMs, obtainable by, for example, running benchmarks on labelled in-domain datasets.

## 3   Methodology

Suppose we have a set of $N$ questions and $J$ LLMs. For each question $q^{(i)}$, we sample $K$ answers from each LLM, resulting in $J \times K$ answers $\{a_{11}, \ldots, a_{JK}\}$, and record their probabilities of being generated $\boldsymbol{y}^{(i)} = \{y_{11}, \ldots, y_{JK}\}$, which are provided as `logprob` or can be estimated by sampling black-box models, and are known as $p(\text{answer})$ [17]. For simplicity, we omit the superscript $(i)$ as the parameters are shared across questions.

We introduce a continuous latent variable $\boldsymbol{z} \sim \mathcal{N}(\boldsymbol{\mu}, \boldsymbol{\Sigma})$ of dimension $(J \times K)$ to model the real truthfulness of the answers. The marginal distribution of $z_{jk}$ has shape $z_{jk} \sim \mathcal{N}(\mu_j, \sigma_j)$ and is determined by the reliability of the LLM $j$. The covariance $\boldsymbol{\Sigma}$ is determined by the semantic relations between answers. Following [12], we posit that semantically similar answers would have correlated truthfulness, while contradictory answers would have negating truthfulness. As such, when considering all $z_{jk}$ jointly, $\boldsymbol{z}$'s correlation matrix $\boldsymbol{R}$ is determined by an NLI model [35] that classifies a pair of answers as entailment, contradiction, or neutral:

$$E_{jk,j'k'} = \text{Entails}(a_{jk}, a_{j'k'}) \quad C_{jk,j'k'} = \text{Contradicts}(a_{jk}, a_{j'k'})$$
$$R_{jk,j'k'} = (E_{jk,j'k'} + E_{j'k',jk})/2 - (C_{jk,j'k'} + C_{j'k',jk})/2$$

which ensures a correlation of 1 for equivalent answers and $-1$ for mutually exclusive ones. The covariance matrix $\boldsymbol{\Sigma}$ is then computed as $\boldsymbol{\Sigma} = \boldsymbol{R} \odot \boldsymbol{\sigma}\boldsymbol{\sigma}^T$ and is projected to the nearest positive semi-definite matrix in the Frobenius norm [2].

The data $\boldsymbol{y}$ is assumed to the noisy and uncalibrated observation of the truthfulness $\boldsymbol{z}$, which is calibrated by logistic regression $y_{jk} \sim \text{Bernoulli}(\text{sigmoid}(\beta_{0j} + \beta_{1j} \cdot z_{jk}))$ such that $\beta_1 \geq 0$. The parameters $\boldsymbol{\mu}$, $\boldsymbol{\sigma}$, and $\boldsymbol{\beta}$ are unique for each LLM and are shared across questions. An example of the full generative model is shown in Fig. 1.

For the ease of implementation, we use EM algorithm to learn the parameters by maximizing log-likelihood using stochastic gradient descent and sampling from $p(\boldsymbol{z}|\boldsymbol{y})$ using NUTS [11, 33, 1]. As with most baseline methods to be compared in Section 4, we select the best answer with the lowest uncertainty as the final answer for TD. For our model, the uncertainty of answer $a_{jk}$ is quantified by the lower bound of 99.9% confidence interval of the mean of the posterior distribution $p(z_{jk}|\boldsymbol{y})$.

## 4   Experiments and Results

**Baselines.**   We perform a comprehensive comparison of our Bayesian approach against recent *statistical* and *agentic* methods for UQ and TD related tasks. In addition, we include *oracle* baselines that always choose answers from a single LLM or choose a correct answers if one exists.

Statistical methods are based on the frequency or likelihood of candidate answers and direct consistency measures, with the most commonplace method being **simple majority voting**. [17] assumes the uncertainty of an answer to be the complement of the probability of it being generated, denoted as **p(answer)**. **Semantic entropy** [20] clusters candidate answers based on semantic similarity and assigns uncertainty based on the entropy of the clusters. **Lin et al. (2024)** [24] provides a graph

Laplacian-based remedy to semantic entropy when $p(\text{answer})$ is not available. The **random** baseline assigns uniform uncertainty to all candidate answers.

**AbstainQA** [8] provides two agentic methods for UQ. The **cooperate** method asks LLMs to provide feedback on candidate answers, which is then summarized by a judge for a true/false judgment. The **compete** method asks LLMs to draft paragraphs supporting alternative answers and reconsider the question, and the uncertainty is estimated by the consistency of final answers. **Debate** [6] is a multi-round debating framework where LLMs refine their answers based on explanations from other LLMs. An LLM judge selects the final answer based on the responses from the last round, which does not provide UQ and is only applicable to TD.

**Tasks and evaluation metrics.** Experiments are done on three open-domain question answering datasets pertaining to different levels of aleatoric and epistemic uncertainty [14]: FreebaseQA [16], AmbigQA [28], and IMDB-Torso [39], with FreebaseQA downsampled to match the size of the other datasets of around 1k. Using 10 different seeds, we sample a total of 40 candidate answers per question from four LLMs: Gemma-2-9B [40], GPT-3.5 [29], Llama-3-8B [7], and Mistral-7B [15].

Given a question and its candidate answers, the goal of UQ is to estimate the uncertainty of the candidate answers and it is evaluated by UQ using the area under the receiver operating characteristic curve (AUROC) that measures how well the uncertainty of the correct answer is separated from the uncertainty of the incorrect answers (Table 1). The goal of TD is to select a correct answer (if any) from the candidate answers and it is evaluated by the final question-answering accuracy (Table 2).

Table 1: Bayesian network outperforms baseline methods on all datasets in terms of AUROC for uncertainty quantification. Top non-oracle results are highlighted in green in Tables 1 and 2.

|  |  | FreebaseQA | AmbigQA | IMDB-Torso |
|---|---|---|---|---|
| Agentic | AbstainQA [8] (cooperate) | 0.57 | 0.56 | 0.52 |
|  | AbstainQA [8] (compete) | 0.77 | 0.71 | 0.69 |
| Statistical | Random | 0.50 | 0.50 | 0.50 |
|  | $p(\text{True})$ [17] | 0.74 | 0.64 | 0.65 |
|  | Simple majority voting | 0.90 | 0.76 | 0.86 |
|  | Semantic entropy [20] | 0.85 | 0.74 | 0.86 |
|  | Lin et al. (2024) [24] | 0.53 | 0.52 | 0.53 |
|  | **Bayesian network (ours)** | 0.93 | 0.82 | 0.88 |

Table 2: Bayesian network has high accuracy when picking a single correct answer for AmbigQA and IMDB-Torso questions, compared to other statistical methods.

|  |  | FreebaseQA | AmbigQA | IMDB-Torso |
|---|---|---|---|---|
| Oracle | Gemma-2-9B | 83.1 | 57.7 | 36.0 |
|  | GPT-3.5 | 93.0 | 72.1 | 52.2 |
|  | Llama-3-8B | 78.0 | 46.5 | 28.7 |
|  | Mistral-7B | 81.3 | 52.2 | 39.6 |
|  | Best answer | 98.0 | 91.0 | 73.7 |
| Agentic | AbstainQA [8] (cooperate) | 87.7 | 59.8 | 41.6 |
|  | AbstainQA [8] (compete) | 86.3 | 60.5 | 41.9 |
|  | Debate [6] | 88.0 | 60.5 | 42.3 |
| Statistical | Random | 83.4 | 61.1 | 42.7 |
|  | $p(\text{True})$ [17] | 84.1 | 54.5 | 39.8 |
|  | Simple majority voting | 91.1 | 67.0 | 49.4 |
|  | Semantic entropy [20] | 92.3 | 70.4 | 50.3 |
|  | Lin et al. (2024) [24] | 81.9 | 56.7 | 36.7 |
|  | **Bayesian network (ours)** | 91.0 | 74.0 | 57.3 |

# 5 Discussion

Our Bayesian model consistently performs the best among all baseline methods when ranking candidate answers to align with truthfulness, making it a promising approach for uncertainty quantification. For improving question answering (TD), the Bayesian model outperforms other statistical models on harder datasets. Despite the relative success of multi-agent models in TD, it is noteworthy that the fast growth of context length and computation constrains scaling with the number of agents, which is crucial to performance [22]. In the meantime, the high computational cost of multi-agent models does not always translate to better performance, as shown in our experiments.

The relative high gap between the best performing model and the "best answer" oracle suggests that relying solely on "wisdom of the crowd" is not always reliable, and the best answer is not always the most popular one [19]. However, our method outperforms other voting and consistency-based methods, especially when the majority of the answers are incorrect. Other methods work at the instance level, while our Bayesian model leverages the information across all questions and answers to learn about the relative reliability of each LLM. This is particularly useful, as the Bayesian model could put more weight on the more reliable LLMs even if they are in the minority, which is not explicitly modeled with other methods.

The answers to FreebaseQA questions are more uniform than those to AmbigQA and IMDB-Torso questions. When the answers are diverse (a signal of difficulty), the Bayesian model outperforms other methods. Uniformity can be measured without ground truths by counting the number of unique answers, and it could be used in the future to determine whether the Bayesian model should be used for a particular dataset.

Although we are exploring methods to rank answers without ground truths, we observe from the oracles that unsurprisingly, having labelled data for benchmarking and calibration might be more beneficial in practical applications. For example, evidence from the oracles and our calibration model ($\beta_1 = 1.3$ for GPT-3.5 and $0.5 - 0.7$ for other LLMs) both show that GPT-3.5 is more reliable and confident than other LLMs, an important clue that current methods have not fully leveraged.

In the typical crowdsourcing setting, annotators work on instances of the same task, while in question answering the questions can be heterogeneous and diverse, which our current model fail to consider. The Bayesian framework allows for the incorporation of factors such as question domain or difficulty, which could be added in the future to improve our model's performance.

# 6 Conclusion

Using a Bayesian generative model, we propose a novel approach to measure the uncertainty of answers from multiple LLM-based QA systems, that would better support informed decisions for abstaining from false answers and identifying truthful ones. As a future direction, the Bayesian model provides a principled method to provide a signal of correctness without ground truths, which could be used for data synthesis to improve question answering systems, in combination with filtering and finetuning, or preference optimization.

## Acknowledgments and Disclosure of Funding

We acknowledge the support of the Natural Sciences and Engineering Research Council of Canada (NSERC). This work is supported in part by a gift from Scotiabank, and is funded in part by the HORIZON Research and Innovation Action 101135576 INTEND "Intent-based data operation in the computing continuum". Jerin George Mathew is financed by the Italian National PhD Program in AI. Dr. Xuefei Ning contributed substantially to the camera-ready version by correcting critical methodological misunderstandings, assisting with method development, and contributing to both experiment design and paper writing.

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
