# OpenReview forum: "A Bayesian Approach Towards Crowdsourcing the Truths from LLMs"
_NeurIPS.cc/2024/Workshop/BDU — NeurIPS BDU Workshop 2024 Poster_

### Official Review · Reviewer_7hQv · 2024-09-23
**Using Crowdsourcing in a new domain**

**Rating:** 8
**Confidence:** 3

**Review:**

This paper introduces a promising Bayesian crowdsourcing approach to improve truth discovery from LLM outputs. It provides a solid foundation for addressing the problems of LLM hallucinations.

Pros:
1. The paper introduces a new method and uses a combination of LLMs outputs using this crowdsourcing approach.
2. The paper illustrates the performance improvements in multiple domains.
3. It examines practical applications of the proposed method in detail and accounts for scalability concerns.

Cons:
1. Might be good for a general audience to understand more about the practical cost involved and what the effect is in real world applications.
2. While the information is presented well, further exploration and expansion of the evaluation datasets might serve the reader well.

---

### Official Review · Reviewer_LJKY · 2024-10-08
**Novel and valuable solution (Bayesian approach) to the field of uncertainty quantification and truth discovery for LLMs, which brings the opportunity in real-word application and to solve the real-world problems.**

**Rating:** 6
**Confidence:** 3

**Review:**

- **Overall**:
The paper introduces a Bayesian crowdsourcing solution that combines multiple responses from multiple LLMs, aiding in uncertainty quantification and truth discovery. The proposed method makes a valuable contribution to addressing the hallucination issue. The experimental results are robust, and the experimental settings are reasonable.

- **Pros**:
    - Novelty: The paper introduces a creative method to address the gap in mitigating LLM hallucination issues. Additionally, this method can be applied to a wide range of use cases, from classification to general QA cases, providing significant value to the industry.
    - Experiment Settings: The paper utilizes three datasets and establishes several baselines to validate the performance of the Bayesian method. The overall experimental settings are solid and persuasive.
    - Readability: The paper is well-organized and highly readable.

- **Suggestions**:
    - Explore the performance of the method across a broader range of tasks. The current focus on QA tasks could be expanded to encompass more diverse tasks.
    - Provide an additional explanation about the agentic method mentioned in the paper.
    - Include more evaluation metrics in the experiments to bolster the evidence provided.

---

### Decision · Program_Chairs · 2024-10-09

**Decision:**

Accept (Poster)

**Comment:**

Reviews are all positive, so I recommend accept.